# A Pvr–AP-1–Mmp1 signaling pathway is activated in astrocytes upon traumatic brain injury

Tingting Li[1], Wenwen Shi[1], Margaret S Ho[2,3]*, Yong Q Zhang[1]*

[1]Key Laboratory of Molecular and Developmental Biology, Institute of Genetics and Developmental Biology, University of Chinese Academy of Sciences, Chinese Academy of Sciences, Beijing, China; [2]Institute of Neuroscience, National Yang Ming Chiao Tung University, Taipei, Taiwan; [3]Brain Research Center, National Yang Ming Chiao Tung University, Taipei, Taiwan

## eLife Assessment

This study represents a **valuable** finding on the neuron-glia communication and glial responses to traumatic brain injury (TBI). The data supporting the authors' conclusions on TBI analysis, RNA-seq on FACS sorted astrocytes, genetic analyses on Pvr-JNK/MMP1 are **solid**. However, cellular aspects of the response to TBI, statistical analysis, and molecular links between Pvr-AP1 are incomplete, which could be further strengthened in the future by more rigorous analyses.

*For correspondence:
margaret.ho@nycu.edu.tw
(MSH);
margaret.ho@nycu.edu.tw
(MSH);
yqzhang@genetics.ac.cn (YQZ)

**Abstract** Traumatic brain injury (TBI) caused by external mechanical forces is a major health burden worldwide, but the underlying mechanism in glia remains largely unclear. We report herein that *Drosophila* adults exhibit a defective blood–brain barrier, elevated innate immune responses, and astrocyte swelling upon consecutive strikes with a high-impact trauma device. RNA sequencing (RNA-seq) analysis of these astrocytes revealed upregulated expression of genes encoding PDGF and VEGF receptor-related (Pvr, a receptor tyrosine kinase), adaptor protein complex 1 (AP-1, a transcription factor complex of the c-Jun N-terminal kinase pathway) composed of Jun-related antigen (Jra) and kayak (kay), and matrix metalloproteinase 1 (Mmp1) following TBI. Interestingly, Pvr is both required and sufficient for AP-1 and Mmp1 upregulation, while knockdown of AP-1 expression in the background of Pvr overexpression in astrocytes rescued Mmp1 upregulation upon TBI, indicating that Pvr acts as the upstream receptor for the downstream AP-1–Mmp1 transduction. Moreover, dynamin-associated endocytosis was found to be an important regulatory step in downregulating Pvr signaling. Our results identify a new Pvr–AP-1–Mmp1 signaling pathway in astrocytes in response to TBI, providing potential targets for developing new therapeutic strategies for TBI.

## Introduction

Traumatic brain injury (TBI) is one of the leading causes of adult disability due to limited neurological recovery (*Joy et al., 2019*). About 43% of people discharged after hospitalization for acute TBI develop long-term locomotor disability (*Ma et al., 2014*). TBI triggers primary neural injuries via rapid physical forces that damage axons, glia, and blood vessels (*Chodobski et al., 2011*; *Daneshvar et al., 2015*), followed by diverse secondary damage such as Wallerian axon degeneration (*Büki and Povlishock, 2006*) and blood–brain barrier (BBB) dysfunction (*Chodobski et al., 2011*) in the central nervous system (CNS). The outcome of TBI is affected by various factors, including age and diet (*Byrns et al., 2021*; *Katzenberger et al., 2016*). In addition, glial cells are critical mediators in

response to TBI in the mammalian brain (*Mira et al., 2021*). During TBI, astrocytes become active, a process characterized by swelling, scar tissue formation, and engulfment of neuronal debris (*Liddelow et al., 2017*; *Sofroniew and Vinters, 2010*), while microglia rapidly transform from their active state to phagocytose cell debris (*Davalos et al., 2005*).

*Drosophila melanogaster* is an ideal model organism for discovering new gene networks and investigating the molecular and cellular responses to CNS injury (*Ayaz et al., 2008*; *Bier, 2005*; *Hoopfer et al., 2006*; *MacDonald et al., 2006*). *Drosophila* astrocyte-like glia (henceforth astrocytes), the conserved counterparts for mammalian astrocytes, and ensheathing glia (ENG) constitute the major glial subclasses in the CNS (*Doherty et al., 2009*; *Stork et al., 2014*). A previous study revealed that *Drosophila* ENG perform the phagocytic function by the engulfment receptor Draper, which interacts with downstream signaling components, including ced-6, Shark, Src42a, c-Jun N-terminal kinase (JNK), and matrix metalloproteinase 1 (Mmp1), in the glial engulfment process in response to axotomy (*Awasaki et al., 2006*; *Doherty et al., 2014*; *Lu et al., 2014*; *Lu et al., 2017*; *Purice et al., 2017*). Although ENG function upon axotomy has been well characterized, the response of other types of glia such as astrocytes to neural injury remains unclear.

In mammals, signaling by platelet-derived growth factor (PDGF) directs cell migration, alters cell shape, and plays a role in wound healing (*Hoch and Soriano, 2003*; *Sil et al., 2018*). Meanwhile, vascular endothelial growth factor (VEGF) signaling is critical for cell proliferation and survival, and vascular endothelial cell mitogenesis and permeability (*Apte et al., 2019*; *Sakurai et al., 2005*). Pvr, short for PDGF- and VEGF receptor-related, is the sole *Drosophila* ortholog of the PDGF and VEGF receptor families (*Cho et al., 2002*). Pvr interacts with different signaling molecules in a context-dependent manner. For example, Pvr regulates thorax closure during metamorphosis via the JNK signaling pathway (*Ishimaru et al., 2004*), but the Pvr-mediated wound-induced epidermal cell migration does not depend on JNK (*Wu et al., 2009*). The JNK/activator protein-1 (AP-1) signaling pathway is involved in different cellular processes, including cell migration (*Pastor-Pareja et al., 2004*), organ size control (*Willsey et al., 2016*), and cell death (*Moreno et al., 2002*). The JNK effector AP-1 complex, composed of heterodimeric Jun/Jun-related antigen (Jra) and Fos/kayak (kay), promotes transcription of genes including Mmp1 that functions in tissue remodeling, regulation of inflammatory processes, and metastasis of cancer cells (*Egeblad and Werb, 2002*; *Page-McCaw et al., 2007*; *Parks et al., 2004*).

Previous studies have implicated a role for ENG, Draper pathway, and the innate immune pathway in response to TBI (*Doherty et al., 2009*; *Lu et al., 2017*; *Purice et al., 2017*; *Anonymous, 2013*). However, whether astrocytes also play a role in TBI response remains elusive. Herein, using a *Drosophila* TBI model induced with a high-impact trauma (HIT) device (*Anonymous, 2013*), we found that TBI caused multiple defects including upregulated expression of innate immune genes and astrocyte swelling. To uncover the molecular pathways activated by TBI, we performed RNA sequencing (RNA-seq) analysis of astrocytes sorted by fluorescence-activated cell sorting (FACS) to profile TBI-induced genes. Interestingly, we showed that the expression of Pvr, AP-1, and Mmp1 was upregulated following TBI. Knockdown of Pvr or AP-1 expression in astrocytes rescued TBI-induced Mmp1 upregulation, while Pvr overexpression promoted Mmp1 expression via AP-1 upon TBI. Our findings identify a new Pvr–AP-1–Mmp1 signaling pathway that acts predominantly in astrocytes in response to TBI, and shed light on the mechanisms of astrocyte responses following TBI.

## Results

### TBI is induced in *Drosophila* adults by a HIT device

To analyze how brains respond to TBI, we applied consecutive strikes to adult flies using a HIT device to induce brain injury as described previously (*Katzenberger et al., 2015*). To quantify the degree of injury, we analyzed the mortality index at 24 hr after TBI ($MI_{24}$), defined as the percentage of dead over the total flies tested. Consistent with a previous report (*Anonymous, 2013*), $MI_{24}$ of 3-day-old adult flies was 20 ± 3.18% after four HIT strikes (*Figure 1A*). $MI_{24}$ after six strikes was 71.88 ± 7.44%, suggesting that increasing the number of strikes results in a higher $MI_{24}$ rate (*Figure 1A*). Given that TBI has been shown to disrupt the formation of the BBB in mammals and flies and blood–eye barrier (BEB) in flies (*Rodríguez-Baeza et al., 2003*; *Sangiorgi et al., 2013*; *Katzenberger et al., 2015*; *Saikumar et al., 2020*), we examined BBB/BEB integrity 2 hr after injecting tetramethylrhodamine-conjugated

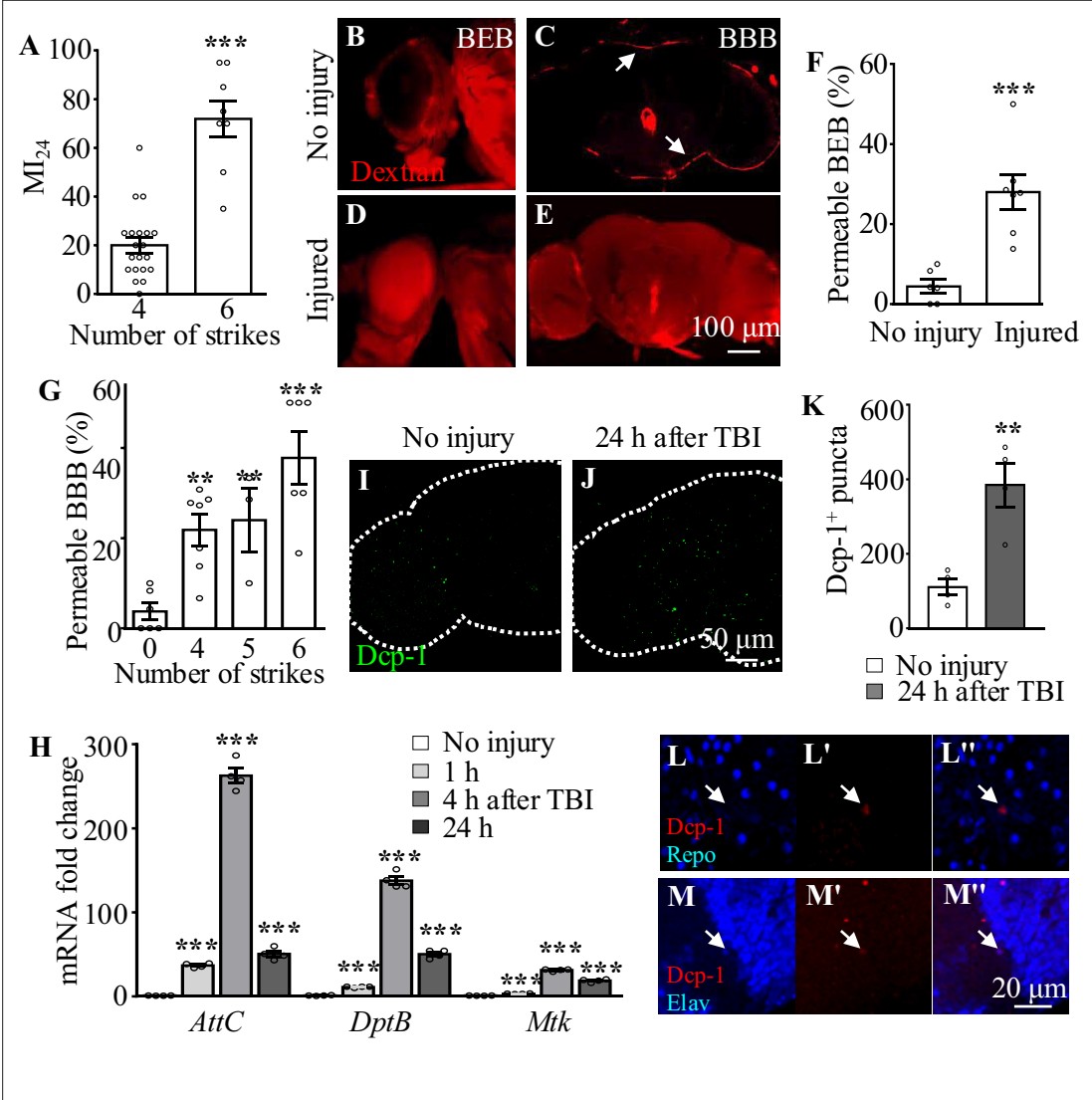

**Figure 1.** Traumatic brain injury (TBI) is induced in adult flies by a high-impact trauma (HIT) device. (**A**) $MI_{24}$ is the mortality index calculated from the percentage of dead over total flies. Increasing the number of strikes from 4 to 6 increases the $MI_{24}$ value. Error bars denote standard error of the mean (SEM; ***$p<0.001$, Student's $t$-test). (**B–E**) The intactness of blood–eye barrier (BEB) and blood–brain barrier (BBB) before or after six strikes is indicated by the accumulated fluorescence intensity at the border of the eye (**B, D**) and brain (**C, E**), respectively. The examination was carried out at 2 hr after fluorescent dye injection. Arrows indicate the hemolymph exclusion line. Scale bar = 100 μm. (**F**) The percentage of permeable BEB with or without TBI (***$p<0.001$). (**G**) The percentage of permeable BBB following different numbers of strikes. Error bars denote SEM (**$p <0.01$ and ***$p <0.001$, Student's $t$-test). (**H**) Histogram showing mRNA expression levels of innate immune response genes of the AMP family (*Attc*, *DiptB*, and *Mtk*) in the adult brain. Expression was increased in all cases following treatment. The *RPL28* gene served as an internal reference. Results are means ± SEM (***$p <0.001$, Student's $t$-test). (**I, J**) The number of Dcp-1⁺ puncta (green) increased in injured adult brains 24 hr after TBI. Scale bar = 50 μm. (**K**) Quantification of Dcp-1⁺ puncta. Error bars denote SEM (**$p<0.01$, Student's $t$-tests). Results are means ± SEM. (**L–M''**) Apoptotic debris labeled with anti-Dcp-1 antibodies (Red). Glial and neuronal nuclei are labeled with anti-Repo (**L**) and anti-Elav (**M**) antibodies (blue), respectively. Scale bar = 20 μm.

dextran (molecular weight = 10,000 Da) into adult fly abdominal segments (*Bainton et al., 2005*). The dye crossed the barrier and penetrated the retina and brain, indicating that both BBB and BEB were disrupted after HIT strikes (*Figure 1B–G*). The percentage of flies exhibiting dextran-permeable BBB increased as the number of strikes increased from 0 to 6, with six strikes resulting in severe BBB disruption (*Figure 1G*) but maintaining a moderate viability providing enough samples for phenotypic analysis. Based on these results, six HIT strikes administered at 5 min intervals were applied to induce TBI throughout the present study.

In addition to disruption of BBB/BEB integrity, we also analyzed TBI-induced secondary damage such as local and systemic immune responses. Given that Toll and immune deficiency (Imd) are the two crucial innate immune response pathways in *Drosophila* (*Valanne et al., 2011*), and their common downstream genes encode antimicrobial peptides (AMPs) that protect flies from TBI injury (*van Alphen et al., 2022*), we analyzed the expression of AMPs in adult fly brains at different time points after TBI. As shown by the real-time reverse transcriptase-polymerase chain reaction (RT-qPCR) results, expression of the AMP genes *AttC*, *DiptB*, and *Mtk* was significantly upregulated at 1 hr and reached a peak at 4 hr after injury (*Figure 1H*), consistent with previous reports (*Barekat et al., 2016*; *Anonymous, 2013*). Taken together, these results suggest that HIT strikes induce TBI reflected by BBB/BEB disruption and elevated expression of innate immune response genes.

## TBI induces neuronal but not glial apoptosis in adult fly brains

Given that neurons and glia have been shown to undergo cell death during both primary and secondary injury responses upon TBI in humans (*Ng and Lee, 2019*), we performed immunostaining with an antibody against Death caspase-1 (Dcp-1), the homolog of human caspase-3, to detect apoptotic cells

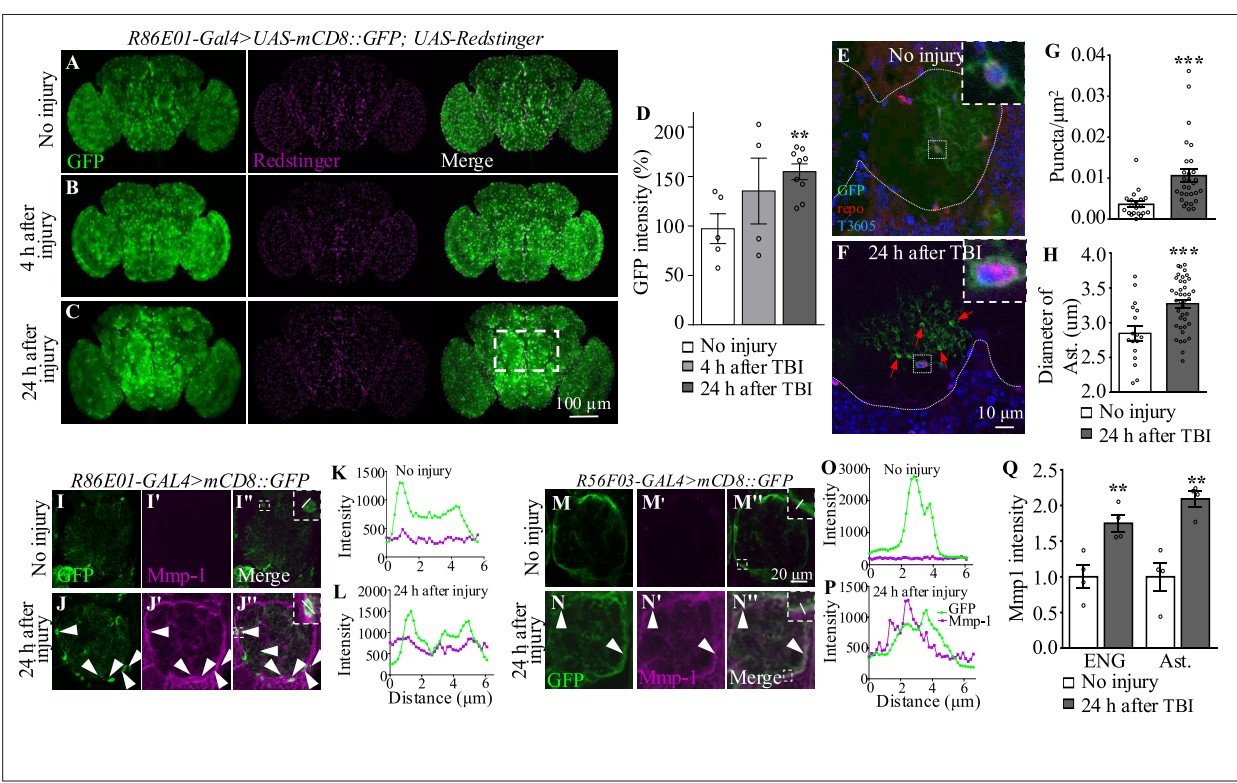

**Figure 2.** Traumatic brain injury (TBI) alters glial morphology and upregulates Mmp1 expression. (**A–C**) Representative images of astrocyte membranes labeled with *mCD8::GFP* driven by the astrocyte driver *R86E01-Gal4*. Co-expression of *UAS-Redstinger* labels astrocyte nuclei. GFP intensities are increased at 4 hr (**B**) and 24 hr (**C**) after injury. The rectangle defined by white dash lines denotes antennal lobes. Scale bars = 100 μm. (**D**) Quantification of GFP intensity at different timepoints after TBI. Results are means ± SEM (**p<0.01, Student's *t*-test). (**E, F**) MARCM (y w UAS-CD8::GFP hs-FLP; tubP-Gal80 FRT-40A/CyO; alrm-Gal4 UAS-CD8::GFP UAS-Dcr2/TM6 Tb Hu cross with *FRT-40A* flies) analysis reveals single astrocytes with evenly distributed GFP⁺ cellular processes without injury (**E**) and an increased number of GFP⁺ puncta at 24 hr after injury (**F**). Arrows denote GFP⁺ accumulations in astrocyte processes. White dots demarcate neuropils. Anti-Repo antibodies labeled glial nuclei. TO-PRO labeled all nuclei. Insets show the morphology of a glial soma before and after injury. Scale bar = 10 μm. (**G**) Quantification of GFP⁺ puncta in astrocyte processes before and after injury. (**H**) Quantification of the astrocyte soma diameter surrounding antennal lobes (ALs) after injury. Results are means ± SEM (***p<0.001, Student's *t*-test). (**I–J''**, **M–N''**) Representative images of ALs co-labeled with GFP and Mmp1 (magenta) in uninjured and injured adult brains. (**J–J''**) GFP-labeled astrocyte membranes (green) overlapped with Mmp1 (magenta) after injury. (**N–N''**) After injury, Mmp1 co-localizes with GFP-labeled ensheathing glia (ENG) (*R56F03-Gal4*) membranes (green). Arrows show Mmp1-GFP colocalization in (**L''**) and (**P''**). Scale bar = 20 μm. (**K, L**) Graph showing fluorescence intensity per pixel in each channel for the inset in (**I''**) and (**J''**) of the astrocyte membrane and Mmp1 along a line drawn through a cell body. (**O, P**) Graph showing fluorescence intensity per pixel in each channel for the inset in (**M''**) and (**N''**) of the ENG membrane and Mmp1 along a line drawn through the process of ENG. (**Q**) Quantification of Mmp1 intensity in (**I–J''**) and (**M–N''**). Ast is the short of astrocytes. Results are means ± SEM (**p <0.01, Student's *t*-test).

in our TBI model. Interestingly, the number of Dcp-1[+] puncta was significantly increased in adult fly brains at 24 hr after TBI (*Figure 1I and J*), with a 2.47-fold increase compared to the number in uninjured controls (*Figure 1K*). Furthermore, we found that almost all Dcp-1[+] puncta were co-localized with Elav rather than Repo, suggesting that neurons but not glia are apoptotic (*Figure 1L–M''*). These results suggest that, as in mammals, TBI also induces neuronal apoptosis in fruit flies.

## TBI leads to astrocyte activation

In addition to the TBI-induced neuronal apoptosis mentioned above, glial cells also become reactive after nervous system injury (*Burda and Sofroniew, 2014*). It has been shown that ENG phagocytose neuronal debris upon axotomy (*Doherty et al., 2009*), but whether astrocytes respond to TBI remains unknown. To explore the potential role of astrocytes in the TBI response, *R86E01-GAL4* (*Kremer et al., 2017*), an astrocyte-specific driver, was used to express *UAS-mCD8::GFP* to label astrocyte plasma membrane. In uninjured fly brains, GFP intensities in the cell bodies and processes of astrocytes were widely distributed throughout the brain (*Figure 2A and D*). While no significant change in GFP intensity was detected in brains at 4 hr after TBI (*Figure 2B and D*), the GFP intensity was significantly increased and the cell bodies of astrocytes were significantly enlarged at 24 hr after TBI in the central brain region, especially in the antennal lobes (ALs) (*Figure 2C and D*). Of note, these results were not affected by the presence of one or two UAS transgenes as the efficiency of *R86E01-GAL4* in driving the expression of either was comparable as shown by the similar GFP levels (*Figure 2A–C, I, and J*).

In mammals, the hallmarks of astrocyte activation are hypertrophy of astrocytes and upregulated level of the glial fibrillary acidic protein (GFAP) (*Pekny and Pekna, 2004*). Given that there is no GFAP homolog in *Drosophila* (*Doherty et al., 2009*), we examined morphological changes in astrocytes at single-cell resolution following TBI by mosaic analysis with a repressible cell marker (MARCM) on adult fly brains at 3–4 days after eclosion. The cell bodies of these astrocytes were located at the interface between neuropil and cortex regions, or in neuropils, and their cellular processes infiltrated into neuropils (*Figure 2E*). Interestingly, astrocytes in injured brains exhibited bright and distinct GFP[+] accumulations associated with their cellular processes at 24 hr after TBI (*Figure 2F and G*). Quantitatively, the number of these accumulations per square nanometer in the cellular processes was significantly increased at 24 hr after TBI; the increase in the number of accumulations might be responsible for the increase in GFP intensity triggered by TBI (*Figure 2G*). Importantly, the diameter of astrocytes cell bodies was also increased at 24 hr after injury (*Figure 2H*). Taken together, these results show that astrocytes become reactive in adult brains at 24 hr after TBI.

Axotomy induces expression of injury-related proteins such as extracellular proteases the matrix metalloprotease (MMP) (*Purice et al., 2017*). In addition to the morphological changes in astrocytes, we observed upregulated expression of Mmp1 in TBI-treated brains (*Figure 2I–J'', M–N'', and Q*). These Mmp1[+] signals were colocalized with membrane-bound *mCD8::GFP* driven by the astrocytes-specific driver *R86E01-GAL4* or ENG-specific driver *R56F03-GAL4* (*Figure 2K, L, O and P*), with a more dominant expression in ENG. These results suggest that Mmp1 might be produced from both types of glia after TBI.

## Transcriptional profiling of astrocytes in adult flies reveals elevated expression of injury-related genes upon TBI

To further investigate how astrocytes respond to TBI, we analyzed transcriptional changes in astrocytes by RNA-seq (*Figure 3A*). Due to the tiny soma and morphological complexity, the analysis of a purified population of glial cells has been difficult and technically challenging. We have optimized a protocol for purifying astrocytes from adult *Drosophila* of *R86E01-Gal4>UAS-Redstinger* by FACS (see details in 'Materials and methods'). As FACS removes most cell masses and RFP-negative cells, the remaining RFP[+] cells are mostly viable, as shown by MitoTracker staining, and purity was 92.96% (RFP[+] and MitoTracker[+] cells/total cells) in uninjured brains (*Figure 3—figure supplement 1A–C'*). Further analysis of the RNA-seq data showed that the astrocyte markers *Eaat1* (110359 reads) and *alrm* (186413 reads) were highly expressed, while *nSyb*, a neuronal marker, was hardly detected (125 reads) at 4 hr after TBI, verifying the purity of FACS-sorted astrocytes. A principal component analysis (PCA) revealed that the variance between different groups (no injury vs. 4 hr and 24 hr after injury) was greater than the variance within groups (*Figure 3—figure supplement 1D*). From analysis of five

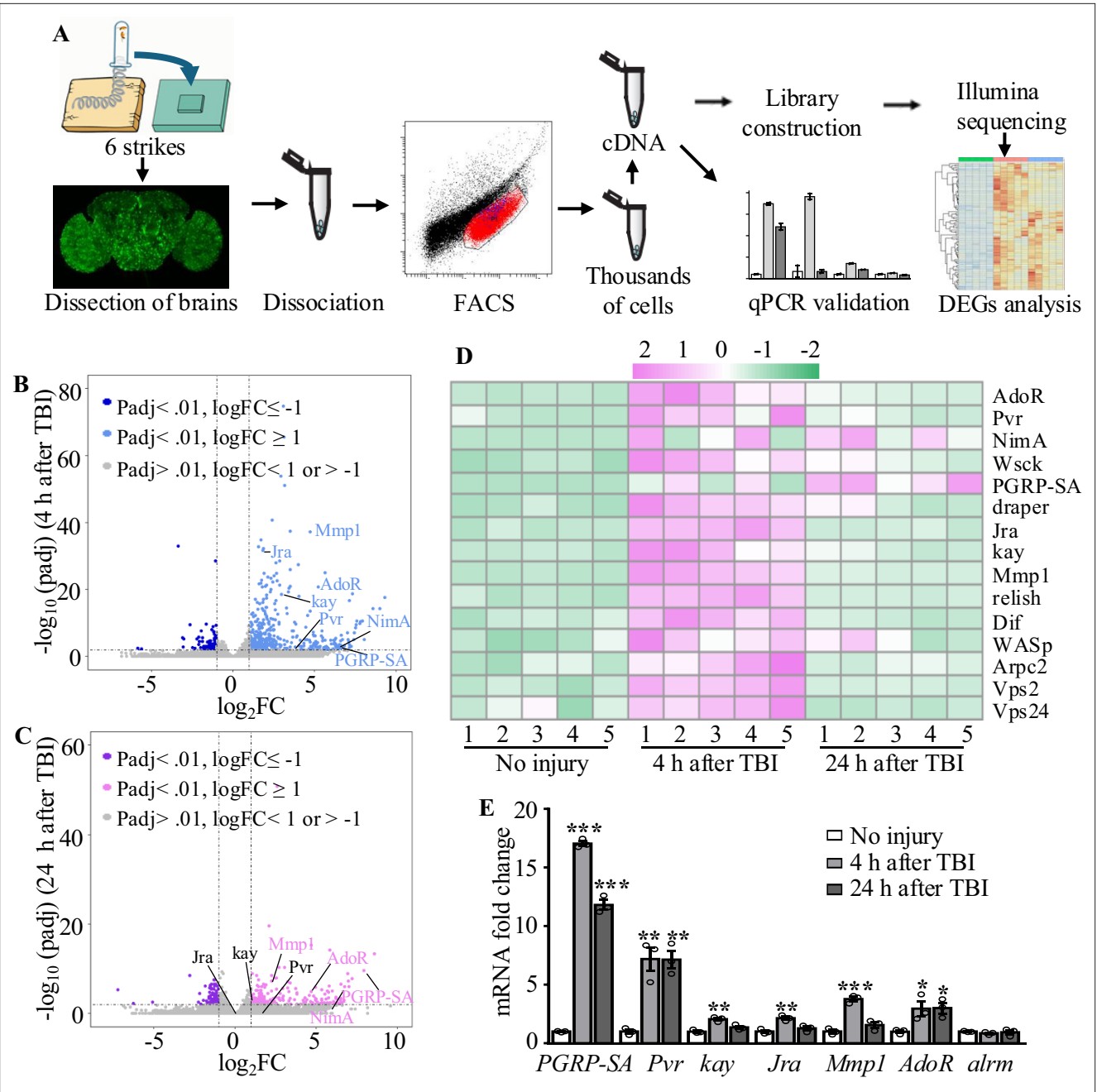

**Figure 3.** Transcriptome analysis of astrocytes reveals upregulation of injury-related and innate immunity genes. (**A**) After dissection, *Drosophila* brains were disrupted and the cell suspension was subjected to fluorescence-activated cell sorting (FACS). Astrocyte cDNAs were amplified by the Smart-seq2 method and a cDNA library was constructed. A subset of upregulated genes from RNA-seq screening was validated by quantitative real-time PCR. (**B, C**) Volcano plots of the gene expression analysis; 522 differentially expressed genes (416 upregulated, 106 downregulated) in astrocytes were detected at 4 hr after injury (**B**), and 357 differentially expressed genes (229 upregulated, 129 downregulated) in astrocytes were detected at 24 hr after injury (**C**). The threshold for differential expression was set at $\log_2 FC = 1$ with an adj. $p<0.01$. FC = fold change; adj. = adjusted (for multiple comparisons by the Benjamin–Hochberg procedure). (**D**) Heatmap of the expression of a small set of representative genes, including *draper*, *Jra*, and *kay*, previously identified as essential factors during glial clearance of severed axons. (**E**) Quantitative real-time PCR validation of a subset of upregulated genes from RNA-seq screening. Error bars represent SEM. Results are means ± SEM (*p<0.05, **p<0.01, and ***p<0.001 by Student's *t*-tests).

The online version of this article includes the following figure supplement(s) for figure 3:

**Figure supplement 1.** Sorting of adult astrocytes by fluorescence-activated cell sorting (FACS).

**Figure supplement 2.** Gene Ontology (GO) analysis of differentially expressed genes at 4 hr and 24 hr after injury.

biological replicates for each time point (no injury, 4 hr, and 24 hr after injury), we obtained 539 million reads (an average of 36 million reads per sample) and aligned the RNA-seq reads to the *Drosophila* genome. We found that 93.67 ± 0.89% of reads mapped to the *Drosophila* genome (dm6 version). Using a cutoff of ≥2-fold differential expression (adj. p-value<0.01, Benjamini–Hochberg procedure), we identified 522 and 357 genes with altered expression at 4 hr (416 upregulated and 106 downregulated) and 24 hr (229 upregulated and 128 downregulated) after injury, respectively (*Figure 3B–D*; *Supplementary file 2*).

Gene Ontology (GO) analysis of these upregulated genes revealed that a significant proportion were associated with endosomal transport, actin cytoskeleton, and wound healing (*Figure 3—figure supplement 2*). Furthermore, we detected upregulated expression of components functioning in the innate immune response after TBI. For instance, expression of the components Relish and Dif of the Toll and Imd pathways was upregulated in astrocytes after TBI (*Figure 3D*), in agreement with a previous finding that Relish is necessary for TBI-induced secondary injuries (*Swanson et al., 2020*). In addition, expression of injury-related genes including *chic*, *Jra*, *kay*, *rl*, *Cad96Ca*, *Idgf4*, *Stam*, *draper*, *Pvr*, *spict*, *Mmp1* and *NijA*, and receptor genes including *Pvr*, *draper*, *AdoR*, *PGRP-SA*, and *NimA* were upregulated in astrocytes at 4 hr or 24 hr after injury (*Supplementary file 3*). A set of endocytic trafficking genes was also identified, including *Arpc1*, *Arpc2*, *cpa*, *cpb*, *Stam*, and *Chmp1*. Subsequent RT-qPCR experiments confirmed the upregulation of all six genes tested at 4 hr and/or 24 hr after TBI (*Figure 3E*), supporting the robustness of the RNA-seq analysis. As a control, the astrocytes-specific *alrm* gene showed no upregulation upon TBI by RT-qPCR and RNA-seq. Taken together, these findings validate the approach of RNA-seq transcription profiling in detecting genes functionally related to injury responses in astrocytes following TBI.

## Pvr and AP-1 are required for the upregulation of Mmp1 after TBI

Among the candidates exhibiting upregulated expression in astrocytes upon TBI, three receptors (Pvr, AdoR, and Draper), all acting upstream of and activating the JNK pathway (*Ishimaru et al., 2004*; *Macdonald et al., 2013*; *Poernbacher and Vincent, 2018*), emerged as new factors with uncharacterized functions in astrocytes (*Figure 3D*). The *Drosophila* RNA interference (RNAi) Screening Center Integrative Ortholog Prediction Tool identified orthologs of these three receptors, subunits of transcriptional factor AP-1 (composed of Jra and kay) of the JNK pathway, Mmp1, and endocytic trafficking genes in mouse and human (*Table 1*; *Hu et al., 2011*). These proteins mediate essential injury responses in vertebrates (*Borea et al., 2016*; *Motz and Coukos, 2013*; *Shi et al., 2021*). To examine a potential role of the receptors and their downstream signaling components in astrocytes after TBI, we quantified the immunostaining intensity of these proteins from projected images of the

**Table 1.** Upregulated genes related to receptors and endocytosis.

| Genes in *Drosophila* | Conserved genes in mouse | Conserved genes in human | 4 hr after injury (FC) | 24 hr after injury (FC) |
|---|---|---|---|---|
| Pvr | pdgfr/vegfr | VEGFR | 13.83 | 2.79 |
| AdoR | Adora2a | ADORA2B | 37.53 | 7.73 |
| Mmp1 | Mmp14 | MMP14 | 26.35 | 4.79 |
| Jra | Jun | JUN | 3.58 | 1.18 |
| kay | FOSL2 | FOS | 6.23 | 1.77 |
| cpa | Capza2 | CAPZA1 | 3.43 | 1.72 |
| cpb | Capzb | CAPZB | 2.01 | 1.25 |
| Stam | Stam2 | STAM | 2.14 | 1.46 |
| Chmp1 | Chmp1b | CHMP1B | 2.16 | 0.92 |
| Vps24 | Chmp3 | CHMP3 | 2.11 | 0.87 |

FC: fold change.

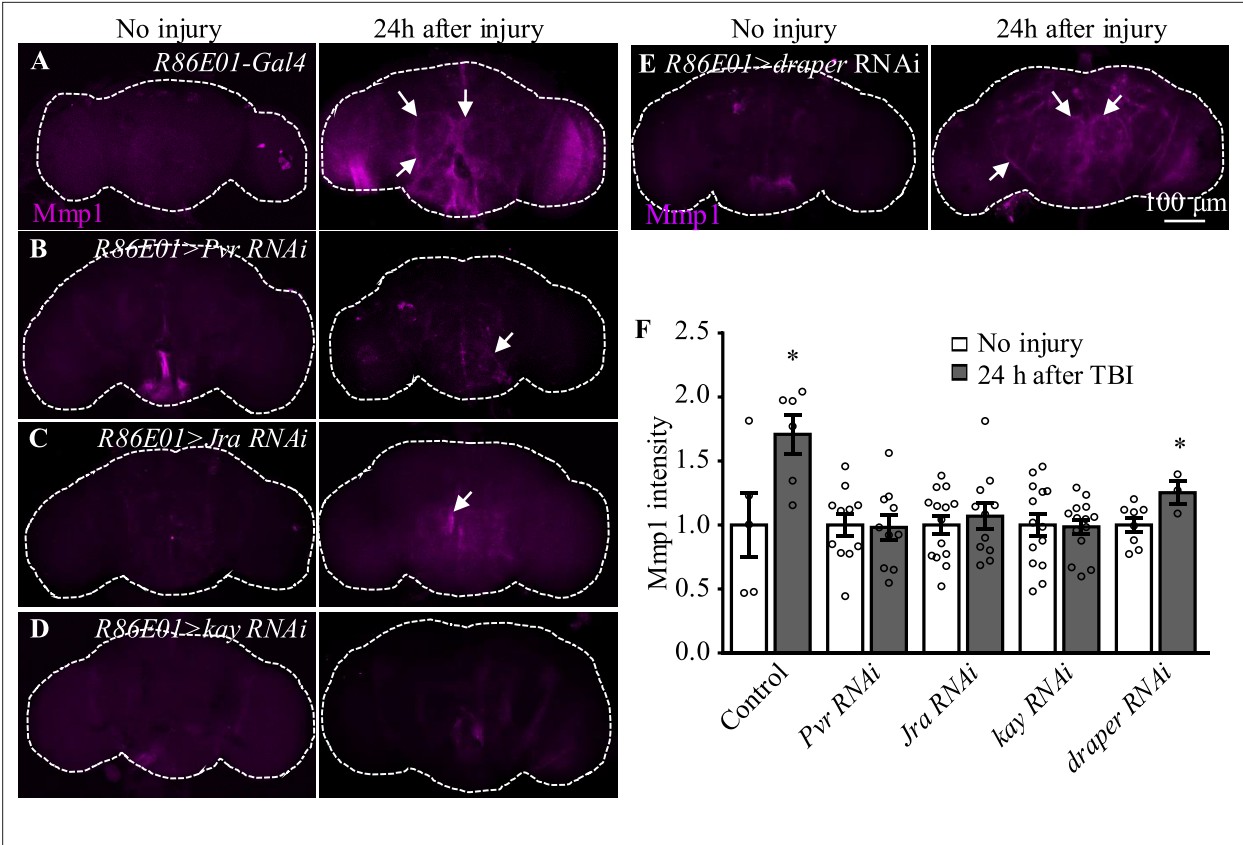

**Figure 4.** Pvr, Jra, and kay are required for Mmp1 upregulation after traumatic brain injury (TBI). (**A–F**) Knockdown of Pvr, Jra, or kay by *R86E01-Gal4*-driven RNAi in astrocytes suppresses Mmp1 upregulation after TBI. Scale bar = 100 µm. All images are projections of 25–30 µm immunostaining slices (one image per 1 µm slice). Arrows show Mmp1 signals. (**F**) Quantification of Mmp1 intensity in different genotypes. Results are means ± SEM (*p<0.05 by Student's *t*-test).

The online version of this article includes the following figure supplement(s) for figure 4:

**Figure supplement 1.** Overexpression of Jra or kay does not enhance the expression of TRE-RFP and Mmp1 in adult brain after traumatic brain injury (TBI).

whole brain (including 20–30 slices with 1 µm per slice). As our RNA-seq analysis revealed elevated expression of JNK-related injury genes such as *AP-1* and *Mmp1* (*Figure 3D*, *Table 1*), we analyzed the expression of Mmp1, a transcriptional target of AP-1 (*Purice et al., 2017*), as a readout for the activity of JNK signaling in response to TBI. We found that upregulated Mmp1 expression induced by TBI was potently suppressed by knocking down the expression of Pvr, Jra, or kay (*Figure 4A–D, F*), indicating that Mmp1 upregulation is mediated by Pvr and AP-1, and predominantly occurs in astrocytes. We note that the location and intensity of Mmp1 signals are variable among different individuals (*Figure 4*), probably because the brain damage caused by HIT is not stereotypical.

Even though flies with Draper knockdown in astrocytes exhibited high mortality, analysis of escapers indicated that Mmp1 upregulation upon TBI was not completely inhibited by Draper knockdown (*Figure 4E and F*). Knockdown of the other receptor AdoR in astrocytes caused a weak inhibition of AP-1 and Mmp1 upregulation (*Figure 4—figure supplement 1A–A'', G–H'', I and J*); *TRE-dsRed* that expresses the red fluorescent protein dsRed under the control of 4×AP-1 motifs (*Chatterjee and Bohmann, 2012*) was used as a reporter for AP-1 activity. As Pvr showed the strongest regulation on Mmp1 expression among the three receptors (*Figure 4*, *Figure 4—figure supplement 1*), we focused on the potential role of Pvr in astrocytes following TBI in the present study. It is noteworthy to mention that although Draper expression was upregulated upon axotomy (*MacDonald et al., 2006*) and TBI (*Figure 3D*), inhibiting Draper expression in astrocytes did not suppress the Mmp1 upregulation after TBI (*Figure 4E and F*). Taken together, these results suggest that Pvr, but not Draper, and AP-1 are required for upregulated Mmp1 in astrocytes after TBI.

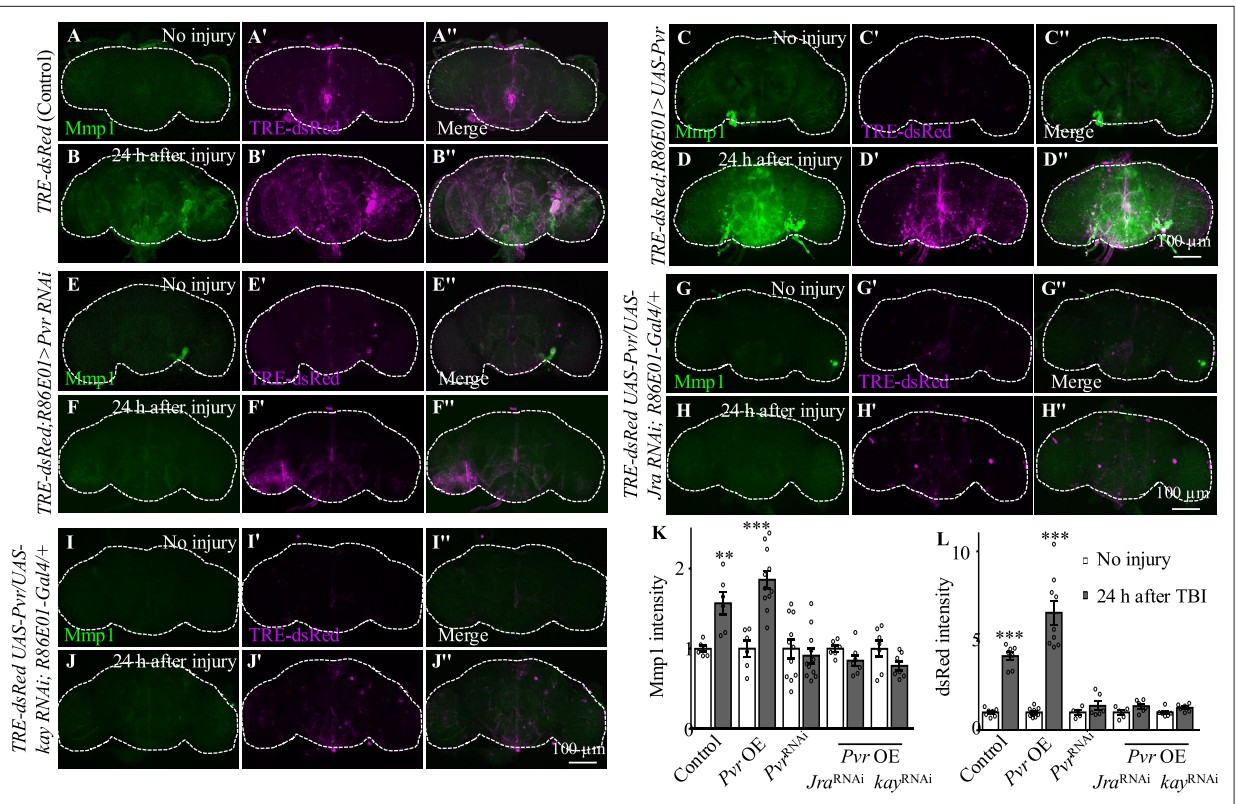

**Figure 5.** Pvr is required for AP-1 upregulation after traumatic brain injury (TBI). (**A–B''**) Adult brains showing increased *TRE-dsRed* (an AP-1 reporter) expression at 24 hr after TBI. (**C–D''**) Overexpression of Pvr in astrocytes leads to upregulated dsREd expression compared with control flies. Scale bar = 100 μm. (**E–F''**) Knockdown of Pvr in astrocytes causes downregulation of dsRed expression compared with control flies after TBI. (**G–H''**) Overexpression of Pvr and knockdown of Jra at the same time in astrocytes causes downregulation of dsRed and Mmp1 after TBI compared with controls. (**I–J''**) Overexpression of Pvr and knockdown of kay in astrocytes at the same time causes downregulation of dsRed and Mmp1 after TBI compared with controls. (**K, L**) Quantification of Mmp1 (**K**) and dsRed (**L**) intensity. Results are means ± SEM. **p<0.01, ***p<0.001, Student's *t*-tests.

## Pvr positively regulates AP-1 signaling and Mmp1 expression in astrocytes after TBI

We next investigated the genetic pathway by which Pvr functions in astrocytes in response to TBI. We examined the activity of AP-1 using the in vivo reporter TRE-dsRed (*Chatterjee and Bohmann, 2012*). In uninjured brains, we detected low TRE-dsRed levels, suggesting a basal AP-1 activity (*Figure 5A–A''*). Similar to Mmp1, TRE-dsRed levels were robustly upregulated at 24 hr after TBI, suggesting activation of JNK/AP-1 signaling (*Figure 5B–B''*). Importantly, TBI-induced expression of TRE-dsRed and Mmp1 was suppressed or further upregulated when inhibiting or overexpressing Pvr expression in astrocytes, respectively (*Figure 5C–F''*). Overexpression of Jra or kay alone did not affect expression of TRE-dsRed or Mmp1 (*Figure 4—figure supplement 1A–F'', I and J*), suggesting that overexpressing a single subunit of the AP-1 complex is not sufficient to upregulate AP-1 activity. While Pvr overexpression caused an increase in TRE-dsRed and Mmp1 expression levels, *Jra* or *kay* RNAi knockdown in the background of Pvr overexpression suppressed the increase (*Figure 5G–L*), indicating that Jra and kay act downstream of Pvr to mediate the injury response upon TBI. These results together show that Pvr is a receptor acting upstream of AP-1 signaling that promotes Mmp1 expression during the injury response in astrocytes (*Figure 5M*).

## Disruption of endocytosis and endocytic trafficking enhances Pvr, AP-1, and Mmp1 upregulation upon TBI

How was the Pvr signaling induced by TBI regulated? It has been shown that VEGFR2 endocytosis proceeds in a clathrin-dependent manner (*Simons et al., 2016*). During clathrin-dependent endocytosis, scission needs Cdc42-interacting protein 4 (Cip4) to recruit dynamin and Wiskott–Aldrich

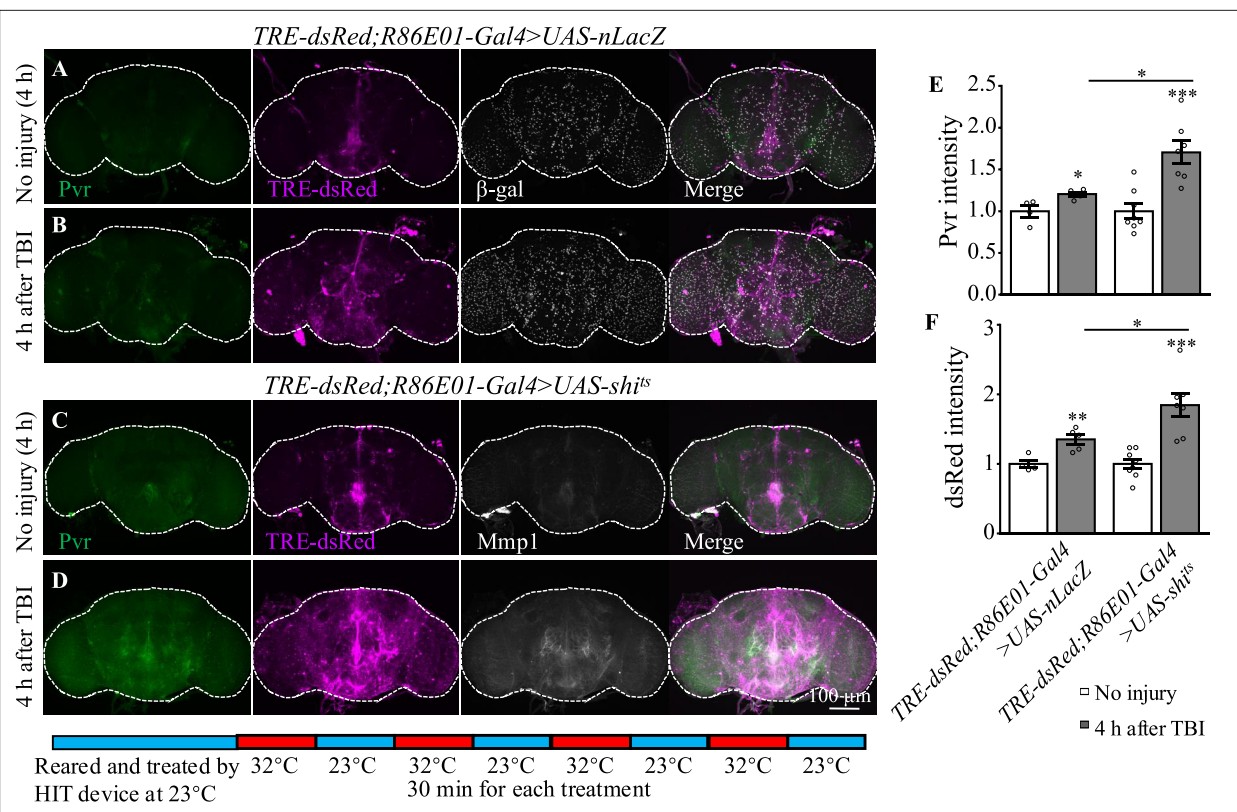

**Figure 6.** Inhibition of dynamin activity increases the expression levels of Pvr and dsRed after traumatic brain injury (TBI). (**A, B**) Representative Pvr immunostaining (green), dsRed (magenta), and β-gal intensity (gray) in control animals (TRE-dsRed;R86E01-Gal4>*UAS-nLacZ*) at 4 hr after TBI. All images are projections of 25–30 µm immunostaining slices (one image per 1 µm slice). (**C, D**) Representative Pvr immunostaining (green), dsRed (magenta), and Mmp1 intensity (gray) in *shi*ʳˢ-expressing animals (TRE-dsRed;R86E01-Gal4>*UAS-shi*ʳˢ) at 4 hr after TBI. For thermogenetic inactivation of dynamin, flies after TBI were incubated at intermittent temperature at 32°C (32°C for 30 min, and then back to 23°C for 30 min, for four rounds, 4 hr in total) until dissection. (**E, F**) Quantification of Pvr and dsRed intensity in different genotypes. Results are means ± SEM (*p<0.05).

syndrome protein (WASp), the activator of the Arp2/3 complex that mediates actin polymerization and promotes the movement of newly formed clathrin-coated vesicles (*Brüser and Bogdan, 2017*). VEGFRs are transmembrane RTKs that are activated when they locate at the cell membrane (*Simons et al., 2016*). We first investigated whether endocytosis is involved in Pvr signaling upon TBI. *Shibire* (*shi*) encodes a GTPase called dynamin that is responsible for fission of the vesicle from the membrane (*van der Bliek and Meyerowitz, 1991*). We analyzed the effect of endocytosis on Pvr signaling by blocking the activity of dynamin with a temperature-sensitive mutant, *shi*ʳˢ, in astrocytes. Interestingly, both Pvr (*Figure 6A, B and E*) and TRE-dsRed (*Figure 6C, D and F*) intensities were increased at 4 hr after TBI when dynamin activity was blocked at the non-permissive 32°C. As a control, expression of nLacZ under the same conditions as mutated dynamin did not exhibit a significant difference in the intensities of Pvr and TRE-dsRed at the permissive 23°C and the non-permissive 32°C after TBI (*Figure 6A, B, E and F*). These results suggest that Pvr signaling is facilitated when endocytosis is blocked.

In addition to Pvr, results from our transcription profiling analysis also revealed upregulation of the expression of actin dynamics and endocytic trafficking genes including *Arpc2*, *cpa*, and *cpb* following TBI (*Supplementary files 1 and 2*). To explore whether endocytic trafficking genes are involved in the injury response in astrocytes, we analyzed Mmp1 and TRE-dsRed expression when inhibiting the expression of Arpc2, cpa, or cpb. Interestingly, TBI-induced Mmp1 levels were further increased after knockdown of each of the three (*Figure 7A–D and O*); consistently, TRE-dsRed expression was also increased after knockdown of *cpa* or *cpb* (*Figure 7H–K and P*). These results indicate that genes involved in endocytosis and endocytic trafficking normally inhibit induction of AP-1 and Mmp1 by TBI in astrocytes.

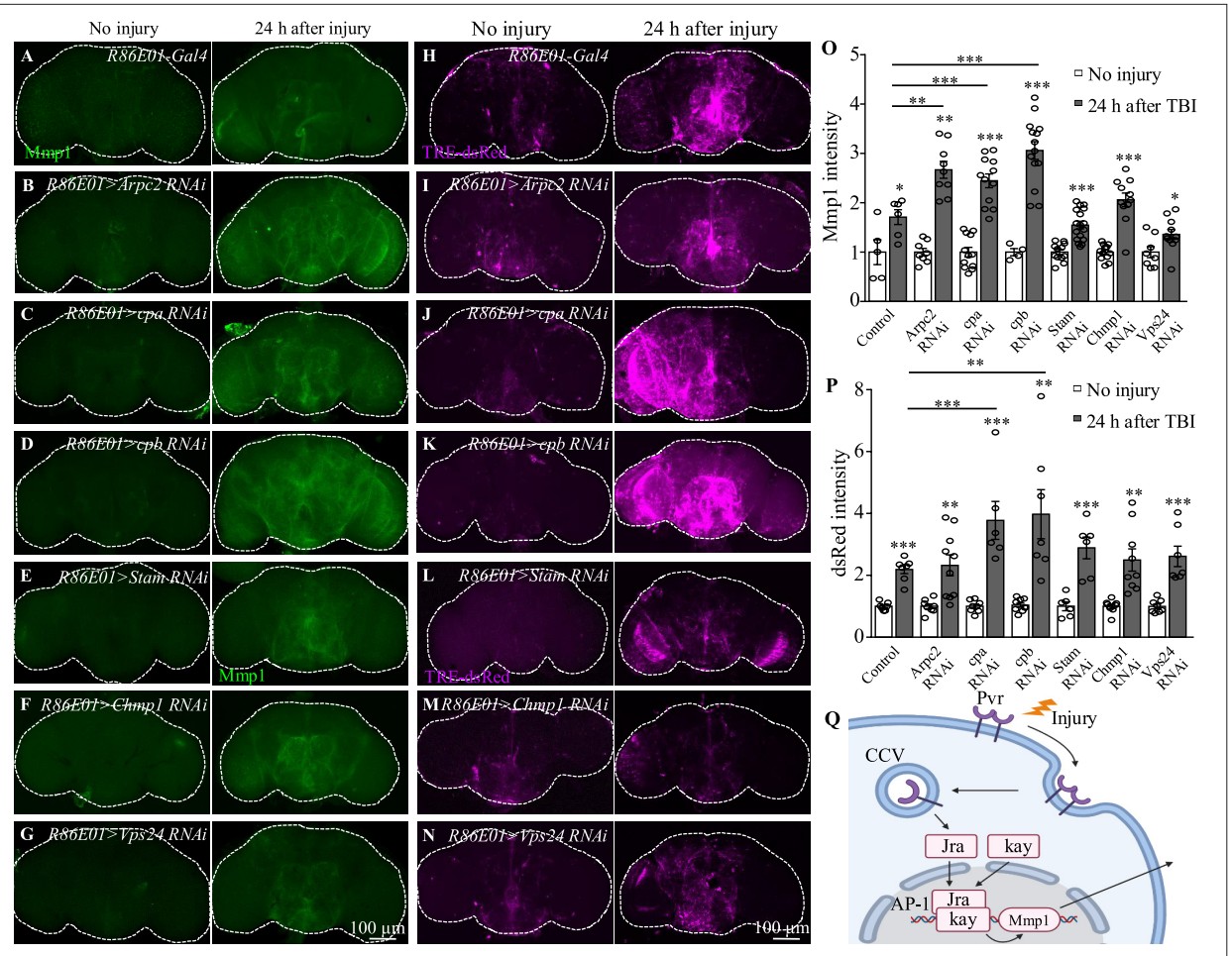

**Figure 7.** Knockdown of endocytic trafficking-related genes up-regulates expression of AP-1 and Mmp1 upon traumatic brain injury (TBI). (**A, H**) Representative Mmp1 immunostaining (magenta) and dsRed intensity (magenta) in control animals (*R86E01-Gal4*) at 24 hr after TBI. (**B–D, I–K**) Knockdown of *Arpc2*, *cpa*, and *cpb* in astrocytes results in further upregulation of Mmp1 and dsRed upon TBI. All images are projections of 25–30 μm immunostaining slices (one image per 1 μm slice). (**E–G, L–N**) Knockdown of *Stam*, *Chmp1*, or *Vps24*, components of endosomal sorting complex required for transport (ESCRT), in astrocytes had no influence on Mmp1 and dsRed upregulation compared with control after TBI (**A, H**). All images are projections of 25–30 μm immunostaining slices (one image per 1 μm slice). (**O, P**) Quantification of Mmp1 and dsRed intensity in different genotypes. Results are means ± SEM (*p<0.05, **p<0.01, ***p<0.001, Student's *t*-tests). (**Q**) Schematic diagram of Pvr–AP-1–MMP1 signaling activated by TBI in astrocytes. AP-1 transcription factor complex is composed of Jra and kay.

In addition, we also observed upregulated expression of a set of endosomal sorting genes, including *Stam*, *Chmp1*, and *Vps24*, that encode components of the endosomal sorting complex required for transport (ESCRT) in astrocytes in response to TBI (*Supplementary files 1 and 2*). These genes are implicated in the formation of intraluminal vesicles (ILVs) from multivesicular bodies (MVBs). Inhibiting expression of the ESCRT-0 subunit Stam and the ESCRT-III subunits Chmp1 or Vps24 in astrocytes however did not result in a significant difference in Mmp1 or TRE-dsRed expression levels compared with controls following TBI (*Figure 7E–G, L–N, O and P*), suggesting that endosome sorting in MVB may not affect Pvr signaling activity.

## Discussion

Using a previously established a *Drosophila* TBI model, we found that TBI causes neuronal apoptosis, BBB/BEB disruption, and astrocyte swelling. Astrocytes respond to TBI by elevated Pvr expression, which in turn enhances the expression of the downstream effectors AP-1 and Mmp1. Our findings identify a new signaling pathway upregulated in astrocytes following TBI.

Unlike vertebrates, in which astrocytes become reactive (reactive astrogliosis) and actively clear neuronal and synaptic debris via phagocytosis upon axon injury (*Liddelow and Barres, 2017*), axon transection in *Drosophila* shows no immediate astrocytic response to injury (*Doherty et al., 2009*; *Purice et al., 2017*). Instead, *Drosophila* ENG express key components of the phagocytic machinery and engulf neuronal debris in response to axotomy (*Doherty et al., 2009*). Here, we examined the molecular and cellular changes of astrocytes after TBI. Upon TBI, astrocytes exhibited apparent damage response including enlarged soma and altered cellular processes containing more membrane-bound GFP$^+$ accumulations (*Figure 2A–J*). These findings suggest that astrocytes are activated upon TBI, demonstrating a critical role for astrocytes in the injury response.

The innate immune response in *Drosophila* relies on the activation of multiple signaling pathways including NF-κB/Relish-mediated Imd and Doral/Dif-mediated Toll signaling pathways. Interestingly, our RNA-seq data revealed elevated expression of genes *Relish* and *Dif* in both pathways upon TBI (*Figure 3D*). However, the expressions of *AMPs* were minimum in astrocytes before TBI and no AMPs genes were identified in differentially expressed genes. Thus, AMPs genes were not activated in astrocytes, though upregulated *AMP* expressions were detected by bulk RNA-seq after TBI. In addition, expression of AP-1 (composed of Jra and kay) and Mmp1 was also upregulated, indicating that the AP-1 signaling pathway is activated in astrocytes following TBI. Even though upregulated Mmp expression in reactive glial responses is an evolutionarily conserved feature of *Drosophila* and mammalian glia (*Cunningham et al., 2005*; *Purice et al., 2017*), transmembrane receptors on astrocytes that regulate Mmp1 expression have not been identified. In mammals, knockdown of VEGF signaling in keratinocytes reduced angiogenesis and delayed wound healing (*Rossiter et al., 2004*). In *Drosophila,* studies of wound healing showed that Pvr likely acts upstream of actin regulatory proteins that promote actin polymerization to lead cells migration (*Brock et al., 2012*; *Wu et al., 2009*).

Through a combination of RNA-seq and genetic analyses, we identified Pvr as a new receptor in astrocytes; Pvr promotes Mmp1 expression via the transcription factor complex AP-1. At present, we do not know how Pvr promotes expression of AP-1 and Mmp1. But previous findings show that Pvr phosphorylates JNK to trigger the downstream signaling containing AP-1 during thorax closure in metamorphosis (*Ishimaru et al., 2004*). During development, the midline glia survival and migration are regulated by the ventral unpaired median and the midline precursor 1 (MP1) midline neurons, the sources of Pvr ligands, that is, PDGF- and VEGF-related factor 1–3 (Pvf1-3) (*Learte et al., 2008*). Additionally, epidermal cell-derived Pvf1 may be sequestered in the blood and exposed upon injury to bind Pvr receptors on wound-edge epidermal cells and initiate the extension of cell processes into the wound gap (*Wu et al., 2009*). Specifically, Pvr ligands bind to Pvr receptor to induce autophosphorylation of tyrosine residues, leading to JNK phosphorylation. Phosphorylated JNK (Bsk) then phosphorylates Jun/Jra transcription factors, triggering Jun/Jra association with Fos/kay, to form the AP-1 complex (*Bogoyevitch and Kobe, 2006*). In addition, glial AP-1 is essential for recovery, ensuring brain integrity and animal survival in the early post-TBI period (*Byrns et al., 2021*). Given that Pvr and AP-1 genetically interact in astrocytes upon TBI, it is likely that Pvr acts upstream of AP-1 and regulates its activity via JNK phosphorylation following brain injury. We speculate that Pvr ligands that may secret from injured cells to activate Pvr-positive astrocytes after TBI; activated Pvr then increases Mmp1 expression via AP-1 and secretion to the intercellular space to participate in injury response.

In addition to Pvr, our RNA-seq data also identified other receptors potentially functioning in astrocytes in response to TBI. Specifically, AdoR expression was upregulated after TBI (*Supplementary file 2*), and knockdown of AdoR expression in astrocytes led to inhibition of AP-1 after TBI (*Figure 4—figure supplement 1G–J*), though at a weaker extent compared with Pvr knockdown. In mammals, A$_{2B}$R, a subtype of AdoR, was shown to stimulate VEGF production in various cell types (*Borea et al., 2016*). Pvr promotes the expression of adenosine deaminase-related growth factor A (Adgf-A), which inactivates extracellular adenosine, the ligand of AdoR (*Zurovec et al., 2002*). Furthermore, Pvr and AdoR collaborate in differentiating cells to maintain the expression of Adgf-A (*Mondal et al., 2011*). Thus, it is possible that Pvr and AdoR act independently or in a coordinated manner to regulate the injury response to TBI in astrocytes in *Drosophila*.

Our RNA-seq profiling also revealed upregulation of Draper. Previous studies revealed a role for Draper-mediated phagocytosis of axonal debris in ENG after axotomy (*Lu et al., 2017*; *MacDonald et al., 2006*); axotomy induces Wallerian-like degeneration (*Büki and Povlishock, 2006*; *Tang-Schomer et al., 2010*). Specifically, damage-associated molecular pattern proteins released from

injured neurons bind to Draper and activate the JNK and JAK/STAT pathways in ENG (*Doherty et al., 2014*; *Freeman, 2015*; *Macdonald et al., 2013*). Here, our findings showed that Draper does not affect Mmp1 expression in astrocytes upon TBI. Although an increase in Draper expression was detected in astrocytes by RNA-seq, knockdown of its expression did not significantly suppress Mmp1 upregulation after TBI. Thus, Draper activates JNK/AP-1 and JAK/STAT signaling in ENG in response to axotomy (*Doherty et al., 2014*; *Macdonald et al., 2013*), whereas astrocytes respond to TBI via Pvr-mediated AP-1 signaling. Despite different pathways being activated in different types of glia upon brain injury, they may cooperate in response to TBI.

Endocytosis and trafficking of VEGFR, the mammalian Pvr homolog, regulate the specificity as well as the duration and strength of the signaling output (*Simons et al., 2016*). Once activated on the plasma membrane and translocated into the cytoplasm, VEGFRs are either shuttled to lysosomes for degradation or recycled back to the plasma membrane to replenish the receptor pool (*Simons et al., 2016*). Our findings showed that disruption of endocytosis by disrupting dynamin activity and inhibiting the expression of Arpc2, cpa, or cpb that regulate endocytic transport into early endosomes enhanced the AP-1 and Mmp1 upregulation in astrocyte following TBI (*Figure 7A–D, H–K, O and P*). As receptor endocytosis attenuates the strength or duration of many signaling processes by physically reducing the concentration of cell surface receptors accessible to the ligand (*Jékely et al., 2005*), blocking dynamin-mediated endocytosis and endocytic trafficking potentially results in accumulated Pvr receptors on the plasma membrane to activate the downstream effector AP-1, thereby enhancing the astrocytes response to TBI. On the other hand, ubiquitinated RTKs are recognized by ESCRT-0, followed by sequential recruitment of ESCRT-I, ESCRT-II, and ESCRT-III onto the membranes of MVBs to form ILVs (*Goh and Sorkin, 2013*). Knockdown of the ESCRT-0 subunit Stam and the ESCRT-III subunits Chmp1 and Vps24 did not affect the expression of Pvr downstream signaling components (*Figure 7E–G, L–N, O and P*), suggesting that Pvr signal transduction was not regulated by endosomal sorting.

Considering the importance of VEGF signaling in a variety of cellular processes, including wound repair (*Folkman and Klagsbrun, 1987*), elucidating the downstream signaling mechanisms, as well as defining the full array of Pvr targets in healthy and diseased brains, will be critical for dissecting the protective or detrimental roles of VEGF in the context of brain injury. Given that the Pvr receptor, AP-1, and the MMP family of secreted proteases are all highly conserved across species, our work provides new insights into how VEGF signaling in glia mechanistically contributes to the TBI response in mammals.

## Materials and methods

### *Drosophila* stocks and husbandry

Flies were maintained on standard cornmeal medium at 25°C otherwise indicated. The following *Drosophila* strains were used: *w1118*, *UAS-mCD8::GFP*, *R86E01-Gal4* (#45914; Bloomington), *R54F03-Gal4* (#39157; Bloomington), *UAS-Redstinger* (#8547; Bloomington), *UAS-Jra* (#7216; Bloomington), *UAS-kay* (#7213; Bloomington), *UAS-Pvr* (#58429; Bloomington), *UAS-nLacZ* (#3955, Bloomington), *PCNA-GFP* (#25749; Bloomington) (*Thacker et al., 2003*), and *TRE-dsRed* (*Chatterjee and Bohmann, 2012*). *UAS-draper-RNAi*, *UAS-Pvr-RNAi*, *UAS-Jra-RNAi*, *UAS-kay-RNAi*, *UAS-Arpc2-RNAi*, *UAS-cpa-RNAi*, *UAS-cpb-RNAi*, *UAS-Stam-RNAi*, *UAS-Vps24-RNAi*, and *UAS-Chmp1-RNAi* flies were obtained from Tsinghua Fly Center (https://thfc.zzbd.org/en). *yw UAS-CD8::GFP hs-FLP; tubP-Gal80 FRT-40A/CyO; alrm-Gal4 UAS-CD8::GFP UAS-Dcr2/TM6 Tb Hu* (alrm-Gal4 is a specific driver in astrocytes [*Doherty et al., 2009*], a gift from Oren Schuldiner) and *FRT-40A* (a gift from Liqun Luo) were used for MARCM analysis of astrocytes. *UAS-shibirets1* (*Pfeiffer et al., 2012*) was provided by Y. Zhong at Tsinghua University.

### TBI assay

Construction and use of the HIT device were described previously (*Anonymous, 2013*). Deflection of the spring to 90° was the standard angle for strike. Mortality index at 24 hr ($MI_{24}$) was determined by analyzing at least 120 flies (20 flies for each test).

### Dye penetration assay

Dye penetration experiments were performed as previously described (*Bainton et al., 2005*). 10 kD Texas red-conjugated dextran (Molecular Probes) at 2.5 mM was injected into the soft tissue between

two abdominal segments of the exoskeleton of flies with and without TBI. Flies were allowed to recover in fresh food tubes and photographed 2 hr later on a Zeiss microscope to detect the integrity of BEB. To detect the integrity of BBB, fly brains were dissected at 2 hr after injection and examined under an Olympus FluoView FV1000 microscope.

## MARCM analysis

MARCM analysis in astrocytes was carried out largely following previous publications (*Jia et al., 2019*; *Lee and Luo, 1999*). Specifically, newly laid eggs were collected for a 4–5 hr interval and cultured at 18°C. Clones were induced by heat shock at 24 hr after larval hatching in a 37°C water bath for 15 min and flies were raised at 25°C after heat shock until immunohistochemistry analysis of adult brain.

## Immunohistochemistry analysis and confocal microscopy

Brain tissues were dissected out from adult after specific treatments. Fixation and staining of adult brains were performed as previously described (*Jia et al., 2019*). Brains were mounted on slides in Vectashield mounting media (Vector Labs). Primary antibodies were used at the following dilutions: chicken anti-GFP (ab13970, Abcam) at 1:1000, mouse anti-Repo (8D12, DSHB) at 1:100, mouse anti-Elav (9F8A9, DSHB) at 1:100, mouse anti-Mmp1 (3A6B4, 3B8D12, and 5H7B11, DSHB) each at 1:20 for all three antibodies, rabbit anti-Pvr (*McDonald et al., 2003*) at 1:1000, rabbit anti-Dcp-1 (Asp216, Cell Signaling Technology [CST]) at 1:100, rabbit anti-pH3 (06–570, Millipore) at 1:1000, and mouse anti-β-gal (40-1a, DSHB) at 1:100. All secondary antibodies (Alexa Fluor 488, 568, or 633 labeled; Molecular Probes) were used at a dilution of 1:1000. The nuclei dye To-Pro-3 Iodide (T3605, Thermo Fisher) was also used at a dilution of 1:1000. The cell-permeant MitoTracker (M22426, Thermo Fisher) was used at a concentration of 100 nM. For soma diameter analysis, we measured the average of long and short axes of astrocytes. All images were collected using an Olympus FV1000 laser scanning confocal microscope and processed with ImageJ. Serial images of 25–30 μm (one image per 1 μm slice) were projected for presentation and statistical analysis.

## Astrocyte preparation for RNA-seq analysis

Flies were collected at 3 days after eclosion and treated with TBI. Brains from adult flies (*R86E01-Gal4/UAS-Redstinger*) with no injury, 4 hr after injury, and 24 hr after injury were dissected out in freshly prepared adult hemolymph saline (AHS) buffer, as previously described (*Yang et al., 2016*). Briefly, a total of ~50 brains for each group was immediately transferred to an Eppendorf tube (1.5 mL) containing 200 μL AHS with Pronase (1 mg/mL; P5147, Sigma-Aldrich) and Dispase (1 mg/mL; LS02104, Worthington Biochemical Corporation). After tissue digestion for 30 min at 25°C, the enzyme medium was removed, followed by two brief, gentle washes with fresh AHS containing 2% PBS. Brain samples were then triturated by gently pipetting in 1 mL AHS through four fire-polished Pasteur pipettes with descending pore sizes (400 μm, 300 μm, 200 μm, and 100 μm) for a total of ~20 min. Next, the dissociated cells were centrifuged for 10 min at $100 \times g$ and the AHS was replaced with the same buffer to minimize any RNA content released from damaged cells during the trituration process. Finally, dissociated tissues were filtered through 45 μm filter and then put on ice. FACS sorting was performed according to the following gating procedure with 70 μm nozzle by BD FACS Aria: (1) an initial SSC-A/FSC-A gate to minimize debris; (2) an SSC-A/pe-Texa Ted-A gate to choose only red fluorescence positive cells. At this step, 1600 cells were sorted directly into ice-cold cell lysis buffer (oligo-dT$_{30}$VN primer, ribonuclease inhibitor, dNTP mix) and stored at –80°C until all samples were ready for RNA extraction and cDNA amplification.

## cDNA library preparation

A total of five biological replicates were collected for each condition (no injury, 4 hr after injury, and 24 hr after injury). Cell samples were sent to the Anoroad Company for RNA extraction. An Oligo-dT primer was introduced to the reverse transcription reaction for first-strand cDNA synthesis, followed by PCR amplification to enrich the cDNA and magbeads purification step was used to purify the products. Briefly, Smart-Seq2 method was used to amplify cDNA. Concentration and integrity of cDNA were assessed using Qubit 3.0 Flurometer (Life Technologies) and Agilent 2100 Bioanalyzer (Agilent Technologies) to ensure the cDNA length was around 1–2 kbp. After quality control, 40 ng cDNA was used to be fragmented at 350 bp by Bioruptor Sonication System (Diagenode Inc). To construct

Illumina library, end repair, 3' ends A-tailing, adaptor ligation, PCR amplification, and library validation were performed. PerkinElmer LabChip GX Touch and Step OnePlus Real-Time PCR System were used for library quality inspection. Qualified libraries were loaded on Illumina Hiseq platform for PE150 sequencing.

## RNA-seq and data analysis

A total of 539 million reads (average 35.9 million reads per sample) was obtained. The fastq data were first processed with Trimmomatic (*Bolger et al., 2014*) to remove Illumina adaptor. The remaining data was aligned against the UCSC *D. melanogaster* genome (dm6) using Hisat2 (*Anonymous, 2013*; *Kim et al., 2019*; *Pertea et al., 2016*) with the annotation file of the same version. On average, 92% of the initial reads were mapped to dm6. HTSeq was used to calculate the raw read counts of genes for each sample. DESeq2 (*Love et al., 2014*) was used to normalize the data. Transcripts were considered as significantly differentially expressed if their adjusted p-values were less than 0.01 ($p < 0.01$) by the Benjamini–Hochberg method. Following normalization of the data, DESeq2 was used to perform pairwise comparisons between the control and experimental groups using parametric tests. Genes with absolute value of logarithmic fold change ($\log_2 FC$) > 1.0 were qualified for the final set of DEGs. Volcano plots and heatmap were constructed for visualization of the resulting expression intensity data following the identification of DEGs and RNA-seq data normalization by the adjusted pvalue calculations.

## Quantitative reverse transcription-PCR analysis

RT-qPCR was performed largely based on previous protocols (*Zhao et al., 2020*). Flies were treated with the same conditions as the RNA-seq experiment.RNA for AMPs analysis was extracted from whole fly brains by TriZol. RNA for confirming RNA-seq findings was isolated from more than 10,000 astrocytes sorted by FACS using the PicoPure RNA isolation kit (KIT0204, Life Technologies). cDNA was generated with SuperScript III First-Strand Synthesis SuperMix (11752250, Invitrogen). For each gene, at least three pairs of qPCR primers were screened by standard curve analysis, and the best primer pairs were chosen based on their efficiency and specificity. Real-time qPCR was carried out by KAPA Library Quantification Kit (KR0389, KAPA Biosystems). Primer sequences are provided in *Supplementary file 1*.

## Thermogenetic inhibition of endocytosis

For inhibition of endocytosis in astrocytes using *UAS-Shi^ts^*, crosses were reared and treated by HIT device at 23°C to avoid unintended inactivation. For thermogenetic inactivation, flies after TBI underwent intermittent temperature treatment at 32°C (32°C for 30 min, and then back to 23°C for 30 min, for four rounds, 4 hr in total) until dissection.

## Statistical analysis

All statistical comparisons were performed using GraphPad Prism software, with significance indicated by ∗∗∗$p \leq 0.001$, ∗∗$p \leq 0.01$, ∗$p \leq 0.05$, and ns, $p > 0.05$. One-way ANOVA was used to compare multiple group means. Student's *t*-tests were used for statistical comparisons between the two groups.

## Acknowledgements

This work was supported in part by the Ministry of Science and Technology of China (2019YFA0707100 and 2021ZD0203900 to YQZ), the National Natural Science Foundation of China (31921002, 31830036, and 31861143031 to YQZ, 32170962 to MH and 31500839 to CX Mao), the Chinese Academy of Sciences Strategic Priority Research Program B grants (XDBS1020100 to YQZ), the 2030 Cross-Generation International Outstanding Young Scholars Program National Science and Technology Council Taiwan (113-2628-B-A49-007 to MH), and Brain Research Center National Yang Ming Chiao Tung University from The Featured Areas Research Center Program within the framework of the Higher Education Sprout Project by the Ministry of Education (MOE) in Taiwan (MH). We thank Bloomington and Tsinghua stock centers for providing fly stocks. We are also grateful to Zhiguo Ma, Oren Schuldiner, Sean Speese, Yong Zhang, Zhiyong Liu, and the members of the Yong Q Zhang lab for discussion and suggestions.

# Additional information

## Competing interests

Margaret S Ho: Reviewing Editor, eLife. The other authors declare that no competing interests exist.

## Funding

| Funder | Grant reference number | Author |
|---|---|---|
| Ministry of Science and Technology of China | 2019YFA0707100 | Yong Q Zhang |
| Ministry of Science and Technology of China | 2021ZD0203900 | Yong Q Zhang |
| National Science Foundation of China | 31921002 | Yong Q Zhang |
| National Science Foundation of China | 31830036 | Yong Q Zhang |
| National Science Foundation of China | 31861143031 | Yong Q Zhang |
| National Science Foundation of China | 32170962 | Margaret S Ho |
| Chinese Academy of Sciences | Strategic Priority Research Program B XDBS1020100 | Yong Q Zhang |
| National Science and Technology Council Taiwan | 113-2628-B-A49-007 | Margaret S Ho |
| Ministry of Education (MOE) in Taiwan | Higher Education Sprout Project | Margaret S Ho |

The funders had no role in study design, data collection and interpretation, or the decision to submit the work for publication.

## Author contributions

Tingting Li, Data curation, Formal analysis, Investigation; Wenwen Shi, Investigation, Methodology; Margaret S Ho, Conceptualization, Formal analysis, Visualization, Writing – review and editing; Yong Q Zhang, Conceptualization, Writing – original draftResources, Writing – review and editing

## Author ORCIDs

Tingting Li ⓘ https://orcid.org/0000-0003-1309-036X
Margaret S Ho ⓘ https://orcid.org/0000-0002-2387-7564
Yong Q Zhang ⓘ https://orcid.org/0000-0003-0581-4882

Reviewer #1 (Public review): https://doi.org/10.7554/eLife.87258.3.sa1
Reviewer #3 (Public review): https://doi.org/10.7554/eLife.87258.3.sa2
Author response https://doi.org/10.7554/eLife.87258.3.sa3

# Additional files

## Supplementary files

• Supplementary file 1. Primer sequences for the RT-qPCR analysis.

• Supplementary file 2. DEGs after TBI.

• Supplementary file 3. Upregulated gene list after TBI.

• MDAR checklist

## Data availability

Data has been deposited to GEO under accession GSE263447.

The following dataset was generated:

| Author(s) | Year | Dataset title | Dataset URL | Database and Identifier |
|---|---|---|---|---|
| Li T, Shi W, Zhang YQ | 2024 | A Pvr-AP-1-Mmp1 signaling pathway is activated in astrocytes upon traumatic brain injury | https://www.ncbi.nlm.nih.gov/geo/query/acc.cgi?acc=GSE263447 | NCBI Gene Expression Omnibus, GSE263447 |

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
