## [Editor Report · eLife Assessment]

This study represents a **valuable** finding on the neuron-glia communication and glial responses to traumatic brain injury (TBI). The data supporting the authors' conclusions on TBI analysis, RNA-seq on FACS sorted astrocytes, genetic analyses on Pvr-JNK/MMP1 are **solid**. However, cellular aspects of the response to TBI, statistical analysis, and molecular links between Pvr-AP1 are incomplete, which could be further strengthened in the future by more rigorous analyses.

---

## [Referee Report · Reviewer #1 (Public review)]

Li et al report that upon traumatic brain injury (TBI), Pvr signalling in astrocytes activates the JNK pathway and up-regulates the expression of the well-known JNK target MMP1. The FACS sort astrocytes, and carry out RNAseq analysis, which identifies pvr as well as genes of the JNK pathway as particularly up-regulated after TBI. They use conventional genetics loss of function, gain of function and epistasis analysis with and without TBI to verify the involvement of the JNK-MMP1 signalling pathway downstream of PVR. They also show that blocking endocytosis prolongs the involvement of this pathway in the TBI response.

The strengths are that multiple experiments are used to demonstrate that TBI in their hands damaged the BBB, induced apoptosis and increased MMP1 levels. The RNAseq analysis on FACS sorted astrocytes is nice and will be valuable to scientists beyond the confines of this paper. The functional genetic analysis is conventional, yet sound, and supports claims of JNK and MMP1 functioning downstream of Pvr in the TBI context.

For this revised version the authors have removed all the unsupported claims. This renders their remaining claims more solid. However, it has resulted in the loss of important cellular aspects of the response to TBI, limiting the scope and value of the work.

The main weakness is that novelty and insight are both rather limited. Others had previously published that both JNK signalling and MMP1 were activated upon injury, in multiple contexts (as well as the articles cited by the authors, they should also see Losada-Perez et al 2021). That Pvr can regulate JNK signalling was also known (Ishimaru et al 2004). The authors claim that the novelty was investigating injury responses in astrocytes in Drosophila. However, others had investigated injury responses by astrocytes in *Drosophila* before. It had been previously shown that astrocytes - defined as the Prospero+ neuropile glia, and also sharing evolutionary features with mammalian NG2 glia - respond to injury both in larval ventral nerve cords and in adult brains, where they proliferate regenerating glia and induce a neurogenic response (Kato et al 2011; Losada-Perez et al 2016; Harrison et al 2021; Simoes et al 2022). The authors argue that the novelty of the work is the investigation of the response of astrocytes to TBI. However, this is of somewhat limited scope. The authors mention that MMP1 regulates tissue remodelling, the inflammatory process and cancer. Exploring these functions further would have been an interesting addition, but the authors did not investigate what consequences the up-regulation of MMP1 after injury has in repair or regeneration processes.

The statistical analysis is incorrect in places, and this could affect the validity of some claims.

Altogether, this is an interesting and valuable addition to the repertoire of articles investigating neuron-glia communication and glial responses to injury in the *Drosophila* central nervous system (CNS). It is good and important to see this research area in *Drosophila* grow. This community together is building a compelling case for using *Drosophila* and its unparalleled powerful genetics to investigate nervous system injury, regeneration and repair, with important implications. Thus, this paper will be of interest to scientists investigating injury responses in the CNS using *Drosophila*, other model organisms (eg mice, fish) and humans.

---

## [Referee Report · Reviewer #3 (Public review)]

In this study, authors used the *Drosophila* model to characterize molecular details underlying traumatic brain injury (TBI). Authors used the transcriptomic analysis of astrocytes collected by FACS sorting of cells derived from *Drosophila* heads following brain injury and identified upregulation of multiple genes, such as Pvr receptor, Jun, Fos, and MMP1. Additional studies identified that Pvr positively activates AP-1 transciption factor (TF) complex consisting of Jun and Fos, of which activation leads to the induction of MMP1. Finally, authors found that disruption of endocytosis and endocytotic trafficking facilitates Pvr signaling and subsequently leads to induction of AP-1 and MMP1.

Overall, this study provides important clues to understanding molecular mechanisms underlying TBI. The identified molecules linked to TBI in astrocytes could be potential targets for developing effective therapeutics. The obtained data from transcriptional profiling of astrocytes will be useful for future follow-up studies. The manuscript is well-organized and easy to read.

However, the connection suggested by the authors between Pvr and AP-1, potentially mediated through the JNK pathway, lacks strong experimental support in my view. It's important to recognize that AP-1 activity is influenced by multiple upstream signaling pathways, not just the JNK pathway, which is the most well-characterized among them. Therefore, assuming that AP-1 transcriptional activity solely reflects the activity of the JNK pathway without additional direct evidence is unwarranted. To strengthen their argument, the study could benefit from direct evidence implicating the JNK pathway in linking Pvr to AP-1. This could be achieved through genetic studies involving mutants or transgenes targeting key components of the JNK pathway, such as Bsk and Hep, the *Drosophila* homologues of JNK and JNKK, respectively. Alternatively, employing p-JNK antibody-based techniques like Western blotting, while considering the potential challenges associated with p-JNK immunohistochemistry, could provide further validation. This important criticism regarding the molecular link between Pvr and AP-1 has been overlooked.

---

## [Author Response]

The following is the authors’ response to the original reviews.

**Recommendations for the authors:**

**Reviewer #1 (Recommendations For The Authors):**
To resolve and further test the claim that TBI did not induce cell proliferation:How many brains did they analyse? Sample sizes must be provided in Figure S1.

As per reviewer’s suggestion, we removed one of the unsupported claims shown in Figure S1. The original Figure S1 is shown below with the sample number added.

The authors could either improve the TBI method or the detection of cells in S-phase, mitosis or cycling. They could use PCNA-GFP or BrdU, EdU or FUCCI instead and at least provide evidence that they can detect cells in S-phase in intact brains. Timing is critical (ie cell cycle is longer than in larvae) so multiple time points should be tested. Or they could use pH3 but test more time points and rather large sample sizes. If they are not able to provide any evidence, then their lack of evidence is no evidence. The authors should consider removing pH3 and PCNA-GFP related claims instead.

We have removed pH3 and PCNA-GFP related results and claims.

Other unsupported claims:Figure 2A-C is not very clear what they are showing, but it is not evidence of astrocyte hypertrophy. It does not have cellular resolution and does not show the cell size, membranes, nor number

(1) We have avoided the term “hypertrophy” and changed the description throughout the text to “astrocyte swelling”.

(2) Images in the resolution of Figure 2E and 2F were able to show the enlarged soma of astrocytes, suggesting swelling.

What is the point of using RedStinger in Figure 2?

We used RedStinger to label the astrocyte nuclei.

Figure S5 is not convincing, as anti-Pvr does not look localised to specific cells. Instead, it looks like uniform background. If they really think the antibody is localised, they should do double stainings with cell type specific markers. If the antibody does not work, then remove the data and the claim. They could test with RNAi knock-down in specific cell types and qRT-PCR which cells express pvr instead.

We have removed the claim that “Pvr is predominantly expressed in astrocytes” and changed the description to “Immunostainings using the anti-Pvr antibodies revealed that endogenous Pvr expression is low in the control brains, yet significantly enhanced upon TBI. Reducing Pvr expression, but not Pvr overexpression, in astrocytes blocked the TBI-induced increase of Pvr expression (Figure S5)”.

Figure S6: it is unclear what they are trying to show, but these data do not demonstrate that astrocytes do not engulf debris after TBI, as there isn't sufficient cellular resolution to make such claim. Firstly, they analyse one single cell per treatment. Secondly, the cell projections are not visible in these images, and therefore engulfment cannot be seen. The authors could remove the claim or visualise whether astrocytes phagocytose debris or not either using clones or with TEM.

We agree with the reviewer that our images do not have the resolution to make this claim. We have removed Figure S6 and corresponding text description.

On statistics:The statistical analysis needs revising as it is wrong in multiple places, eg Fig.1F,G,H; Figure 2D. They only use Student t-tests. These can only be used when data are continuous, distributed uniformly and only two samples are compared; if more than 2 samples, distributed uniformly, then use One-Way ANOVA and multiple comparisons tests. If data are categorical, use Chi-Square.

We have double checked and compared the experimental group to the control separately using the Student t-tests throughout the study.

Other points for improvement:Figure 2E,F: what are GFP puncta and how are they counted?

I. Each GFP puncta looks like a little circle, likely representing a functional or dysfunctional structure. The biology of the GFP puncta is currently unkonwn.

II. We used the ImageJ to quantify the GFP puncta:

(1) Image- type-8 bits

(2) Process-subtract background (Rolling ball radio:10)

(3) Image-Adjust-Threshold-Apply

(4) Analyze-Measure-set measurements-choose “area” “limit to threshold”-OK

(5) Count the puncta number in the choosing area.

(6) Get the number of puncta per square micron.

All genotypes must be provided (including for MARCM clones), currently they are not.

We have shown the full genotype in the corresponding legend.

Figure 7O,P indicate on figure that these are RNAi

We have revised the labels to RNAi in Figure 7O,P.

**Reviewer #2 (Recommendations For The Authors):**
Several typos are present in the text.

We have read the manuscript carefully and corrected typos throughout.